# Pre-Exposure Prophylaxis Interventions among Black Sexual Minority Men: A Systematic Literature Review

**DOI:** 10.3390/ijerph19041934

**Published:** 2022-02-09

**Authors:** Rodman E. Turpin, David J. Hawthorne, Andre D. Rosario

**Affiliations:** 1Department of Epidemiology and Biostatistics, School of Public Health, University of Maryland, College Park, MD 20742, USA; 2Department of Behavioral and Community Health, School of Public Health, University of Maryland, College Park, MD 20742, USA; davidjh@umd.edu; 3Department of Psychiatry and Behavioral Sciences, Howard University Hospital, Washington, DC 20060, USA; arosario@huhosp.org

**Keywords:** LGBTQ, HIV, prevention, experiment, review

## Abstract

Background: Interventions to promote HIV pre-exposure prophylaxis (PrEP) among Black sexual minority men (BSMM) are especially important, given the disproportionate HIV incidence and relatively low uptake of PrEP among BSMM. Methods: We conducted a systematic literature review to identify the characteristics of interventions between 2016 and 2021 promoting PrEP use among BSMM. We synthesized these studies based on sample size, location, the use of peer-based delivery, and key intervention targets. Results: Of the starting total 198 articles, 10 were included in the final review, with the majority of included studies being randomized controlled trials. We identified providing PrEP access, PrEP counseling, HIV and PrEP education, linkage to general health care, and peer-based support as key successful intervention components. The starkest difference between interventions with and without demonstrated PrEP improvements was the outcome: all interventions focused on PrEP initiation led to large improvements, but those focused on PrEP adherence did not. No other factors demonstrated distinct differences between successful and unsuccessful interventions. Conclusion: We identified notable differences in intervention efficacy between PrEP initiation and PrEP adherence outcomes; PrEP adherence is necessary for optimal HIV prevention. Future interventions promoting and measuring PrEP adherence, with a focus on cultural competence and peer components, are recommended.

## 1. Introduction

Black sexual minority men (BSMM) are at the greatest risk of HIV acquisition across race and sexual minority status, accounting for 26% of all new HIV diagnoses in the U.S.A., yet comprising less than 1% of the U.S.A. population [1,2]. Additionally, HIV diagnoses have increased among young adult BSMM (aged 25 to 34) from 2010 to 2015 [1]. Although the incidence rates among BSMM have remained stable since 2015, it is estimated that if the current incidence persists, one in two BSMM will be diagnosed with HIV in their lifetime [2]. For this reason, biomedical approaches to HIV prevention, including the use of HIV Pre-exposure Prophylaxis (PrEP), are important to health equity goals for this population. Despite the well-documented efficacy of PrEP use and the relatively high HIV incidence among BSMM, BSMM PrEP uptake has not been as high as that in White or Hispanic/Latino MSM [2,3,4,5,6,7].

Several studies have identified barriers to PrEP uptake among many populations of SMM [3,5,8]. Barriers to PrEP utilization may include individual factors as well as social and structural barriers. Previous research has identified the following barriers to HIV prevention: difficulties accessing healthcare (e.g., lack of health insurance, geographic distance), infrequent routine healthcare utilization, socioeconomic difficulties, low self-perceived HIV risk, depressive symptoms, and HIV stigma [3,5,6,8,9,10,11]. Social support and resilience have been shown to reduce HIV risk and improve HIV prevention outcomes [4,6,8,9,10,11,12,13,14,15,16,17,18,19,20,21,22,23,24,25,26,27,28,29,30,31,32,33,34]. In totality, there are several relevant factors that affect PrEP use among SMM.

Some of these factors have been important targets in interventions to promote PrEP use among BSMM; research on PrEP interventions in this population are limited, however. Recent studies include interventions utilizing social networks among SMM to improve PrEP initiation and other forms of HIV prevention [5,6,9,10]. Social network approaches may help reduce barriers through fostering a safe environment free from judgment and discrimination, providing greater understanding and acceptance, and improving peer resilience. The use of peer educators is another strategy for PrEP promotion, as these peer educators can effectively disseminate health information within the network and support peer engagement in HIV prevention activities. This type of intervention has already proven successful in HIV stigma reduction efforts [11]. Peer-based interventions can be especially effective at changing norms within social groups, allowing for a sustained intervention impact. Additionally, interventions may directly target the access to PrEP by providing sustained linkage to PrEP services. Systematically synthesizing information from studies on PrEP interventions among BSMM can better inform on what factors are likely to be successful intervenable targets and what approaches to intervention delivery are the most effective.

The purpose of our study was to conduct a systematic literature review of interventions to promote PrEP use among BSMM, a population with demonstrated need for greater PrEP uptake. We will synthesize these studies based on several characteristics, including sample size, location, the use of peer-based delivery, and key intervention targets. We will also examine differences between interventions that were associated with an increase in PrEP use among BSMM and those that were not. We hypothesize that interventions associated with a statistically significant increase in PrEP use will be characterized by having peer-based components and targeting both social and structural barriers to PrEP use.

## 2. Materials and Methods

### 2.1. Search Strategy

A systematic review of the literature was conducted according to the preferred reporting items for systematic reviews and meta-analyses (PRISMA) guidelines [35]. All searches were conducted through the University of Maryland world catalog (WorldCat) search engine. This search engine covers numerous databases subscribed to by the University of Maryland, including PubMed, PsycINFO, and ScienceDirect. We searched for English-language studies published from 2016 through 2021, using the search terms (PrEP OR Pre-Exposure Prophylaxis) AND (Intervention OR Trial OR RCT) AND (Black OR “African American”) AND (MSM OR “Men who have sex with men” OR SMM OR “Sexual minority men”).

### 2.2. Inclusion Criteria

We included studies in peer-reviewed journals that used an intervention design, measured PrEP initiation, referral, or adherence as an outcome of the intervention, and were conducted among a sample consisting of more than 50% Black sexual minority men. Potential intervention designs could include randomized controlled trials or quasi-experimental designs such as single-group pre-test–post-test. Here, sexual minority may include men who have sex with men, or men identifying as a sexual minority (e.g., gay, bisexual, queer). Any study that did not meet all three of these criteria was excluded.

### 2.3. Exposures and Outcome

The outcomes measured for all studies included PrEP initiation, PrEP referrals, and PrEP adherence. The exposures were assignment to the intervention groups. Additionally, we measured which factors were mechanisms of the uptake in PrEP use (e.g., socioeconomic barriers to PrEP use, healthcare access, HIV knowledge, PrEP and related stigma).

### 2.4. Bias Assessment

Bias was assessed using the JBI-MASTARI instruments (Joanna Briggs Institute, Adelaide, Australia). These instruments allow for the calculation of a bias score (ranging from 0 to 100%) based on 8 to 10 questions specific to a given study design. A score of 71% or higher indicated a low risk of bias, of 51–70% indicated moderate risk, and of 50% or less indicated high risk. As our review was limited only to intervention studies, we used the JBI-MASTARI instruments for randomized controlled trials and quasi-experimental studies. Three researchers were involved in the assessment of bias. Each study was assessed for bias by both lead members of the team. In instances where there was a disagreement, the two lead team members met to attempt to come to an agreement. In instances where the two lead team members did not come to an agreement, the third team member would make the final decision on the bias assessment. In all instances, there was ultimately an agreement on the bias assessment.

### 2.5. Data Synthesis

Studies were synthesized (i.e., summarized in aggregate based on key characteristics) based on sample size, years of data collection, study selection method (e.g., hospital/clinic based, community/venue based), location (e.g., U.S.A., other countries), intervention design (e.g., randomized controlled trial, non-randomized), targeted factors, efficacy in increasing PrEP use, and results of the bias assessment. We also assessed the measure of association and statistical significance.

## 3. Results

### 3.1. Search Results and Bias Assessment

Of the starting total 198 articles, 21 were excluded as duplicates, and 158 were excluded based on the abstract (Figure 1). Articles were excluded based on the abstract if it was evident that they were not interventions focused on PrEP. Of the 19 articles reviewed at full text, 10 were included in the final review [36,37,38,39,40,41,42,43,44,45]. The most common reason for studies being excluded after reviewing the full texts was not having a measured PrEP outcome. All studies included had a low potential for bias based on our bias assessments (Appendix A).

### 3.2. Study Designs

The majority of the included studies (6 out of 10) were randomized controlled trials (Table 1). Two of the included studies had single-group pre-test–post-test designs, while one was a non-randomized two-group intervention, where participants could select the PrEP intervention or the comparison group. All included studies were U.S.A.-based, and the majority (9 out of 10) were conducted in major metropolitan cities (Harlem, NY; New York City, NY; Chicago, IL; Los Angeles, CA; Washington, D.C.), with 2 of these studies conducted at multiple sites nationally. Intervention timeframes ranged from 3 months to 1 year. Sample sizes ranged from 31 to 368 after loss to follow-up.

### 3.3. Intervention Mechanisms and Targets

Interventions covered a wide range of PrEP-related content (Table 2). Every single intervention included PrEP access (by providing PrEP through the study) and PrEP counseling (discussing the process of starting and maintaining PrEP, reasons for starting PrEP, and answering participant questions) as components of the intervention. The majority of interventions (7 out of 10) also included a peer-related component, such as a peer navigator or peer-based support groups. Improving PrEP communications (e.g., willingness to discuss PrEP with peers, family, and healthcare workers) was a common focal point. Other commonly utilized intervention targets were PrEP knowledge, HIV knowledge, and HIV/STI risk counseling. Half of the interventions included linkage to general healthcare as a component (e.g., linkage to overall routine healthcare services not limited or related to PrEP). One intervention also included more broad access to socioeconomic resources, including employment, housing, and legal resources. Only two interventions targeted reduction of stigma (e.g., PrEP stigma, sexual stigma) and a mechanism of PrEP promotion.

### 3.4. PrEP-Related Outcomes

Interventions primarily focused on PrEP use/initiation (seven studies) or PrEP adherence (two studies). One study mentioned in our review used PrEP referrals as an outcome. Here, this included referrals to PrEP care from a dedicated PrEP referral line and attendance at a first referred PrEP care clinic appointment. Associations between interventions and outcomes are presented as odds ratios (Table 2). Odds ratios were only included for multi-group interventions or interventions with pre-test and post-test PrEP measures. Odds ratios for single-group interventions restricted to those not using PrEP at baseline could not be calculated. Among the studies measuring PrEP use/initiation at the end of the study period, one study measured PrEP use while sexually active. All but two randomized controlled trials found a significant difference between intervention and control groups. For one of these studies, however, this was likely due to the small sample size (*n* = 61 after loss to follow-up), as there were large observed effect sizes (OR = 3.03). The four interventions with the largest effect estimates (OR > 2.00) included a peer navigation component, though the only intervention with a null effect estimate did as well. This latter study utilized PrEP adherence however, while all the former studies measured PrEP initiation. Overall, interventions measuring PrEP initiation as their outcome demonstrated much larger effect estimates than those measuring PrEP adherence as the outcome. For one intervention measuring PrEP adherence, the effect estimate was null (OR = 1.00). For the other study, this was a single-group intervention with no pre-test–post-test comparisons, though it noted nearly a 50% drop in adherence within 4 weeks of the intervention, and only a third of the sample having detectable PrEP levels by week 48 of the intervention [43]. Additionally, BSMM in this study had the lowest adherence among all the racial/ethnic groups in the study.

## 4. Discussion

We identified several targets for intervention approaches to PrEP promotion among BSMM, with PrEP access and counseling being utilized in all reviewed studies [36,37,38,39,40,41,42,43,44,45]. PrEP knowledge, HIV knowledge, HIV/STI risk counseling, and linkage to general healthcare were commonly observed as well. These factors reflect two general constructs: the understanding of HIV risk and PrEP benefits (reflected in knowledge and counseling) and providing access to PrEP (reflected in both PrEP access and general health linkage). Peer-based interventions were frequently utilized and were a largely efficacious approach to promoting PrEP initiation among BSMM. Overall, focusing on these constructs and utilizing peer-related components appeared to demonstrate promise in increasing PrEP initiation among BSMM, though this efficacy was not observed using adherence outcomes.

We identified a stark difference in PrEP outcome improvement between interventions with a PrEP initiation outcome and those with a PrEP adherence outcome. Every intervention in our review that measured PrEP initiation or referrals as the outcome demonstrated either significant findings or large effect estimates. In contrast, the two studies that measured PrEP adherence as the outcome had largely null findings, in terms of both a lack of statistical significance and of ratio effect estimates very close or equal to 1. No other notable differences in design or intervention targets were observed between interventions with and without successful PrEP outcomes. PrEP initiation appeared to be a much more successfully intervenable outcome than PrEP adherence, though initiation without adherence is not likely to be effective in providing sufficient protection from HIV acquisition. Ensuring that interventions are tailored to promote long-term PrEP adherence rather than short-term initiation/linkage is critical to making the greatest gains in HIV prevention among BSMM. Interpretations of non-adherence to PrEP can be challenging however, as this could reflect the fact that someone is taking PrEP ineffectively or that their personal circumstances no longer warrant PrEP use. Identifying the reason for PrEP cessation is especially important for determining the efficacy of PrEP interventions. One study captured some of these personal circumstances related to PrEP use, as the outcome measured was PrEP use while sexually active. This provides additional context that helps us more effectively interpret the reasons for inconsistency in PrEP use. Future research should investigate this further.

There are several potential challenges to consider in developing PrEP promotion interventions for BSMM. BSMM both historically and currently face intersectional stigma related to both sexual identity and race. Much of this stigma is related to interactions with healthcare providers, which can then be a barrier to healthcare engagement [3,17]. Additionally, stigma broadly can result in greater depression, isolation, and thus marginalization from health messaging [26]. Structural racism can also limit the access to healthcare, such as income-related racism resulting in BSMM having greater difficulty affording PrEP, or housing-related racism creating greater distances from healthcare services [2,3]. Additionally, there is a litany of historical abuses of Black individuals, including BSMM, by the medical system. Taken together, these factors create significant barriers to accessing BSMM and utilizing PrEP. The delivery of PrEP through community-based organizations tailored to and lead by BSMM can help address many of these issues. The common use of peer-based interventions similarly helps in many of the aforementioned issues. Overall, PrEP interventions based on cultural competency and cultural humility are greatly needed.

There are important strengths and limitations to consider in interpreting our findings. First, our review only consisted of 10 studies, so its generalizability is limited. Only 2 of the 10 studies measured PrEP adherence as an outcome; despite this small sample size, the difference between studies measuring PrEP initiation/referrals as outcomes and those measuring PrEP adherence was remarkably stark, with every single intervention measuring PrEP initiation or referrals demonstrating some degree of change in the outcome, while both studies measuring PrEP adherence as the outcome did not. The focus on BSMM also limits the generalizability of our conclusions, though this population is important for PrEP promotion efforts, given its disparately high burden of HIV compared to SMM of other racial/ethnic groups. Finally, publication bias was likely to affect the studies that we could include in our review, though approximately a third of the included multi-group studies had null findings.

## 5. Conclusions

We identified several commonalities in intervention approaches to PrEP promotion among BSMM, including providing PrEP access and counseling. The use of peer-based interventions appeared as a common and effective approach to promoting PrEP initiation among this population. The most successful interventions tended to utilize several multidimensional components, including targeting HIV and PrEP knowledge and linkage to general health care. Based on our findings, however, PrEP initiation appears to be a more successfully intervenable outcome than PrEP adherence. Future intervention work promoting and measuring PrEP adherence is especially important, as long-term adherence to PrEP is necessary for optimal gains in HIV prevention. This has significant implications for the HIV epidemic and its disproportionate burden among BSMM compared to SMM of other racial/ethnic groups. Interventions utilizing novel strategies for promoting consistent adherence to PrEP among BSMM are strongly recommended. We also recommend future research into stigma as a critically important factor in the relationship BSMM have with healthcare services, including utilization of PrEP; internalized stigma in particular may be a relevant intervention target in efforts to promote PrEP uptake among BSMM. This was largely underutilized as an intervention target in our review. Overall, culturally tailored approaches to PrEP promotion among BSMM are a necessary tool for both understanding and combating the HIV epidemic.

## Figures and Tables

**Figure 1 ijerph-19-01934-f001:**
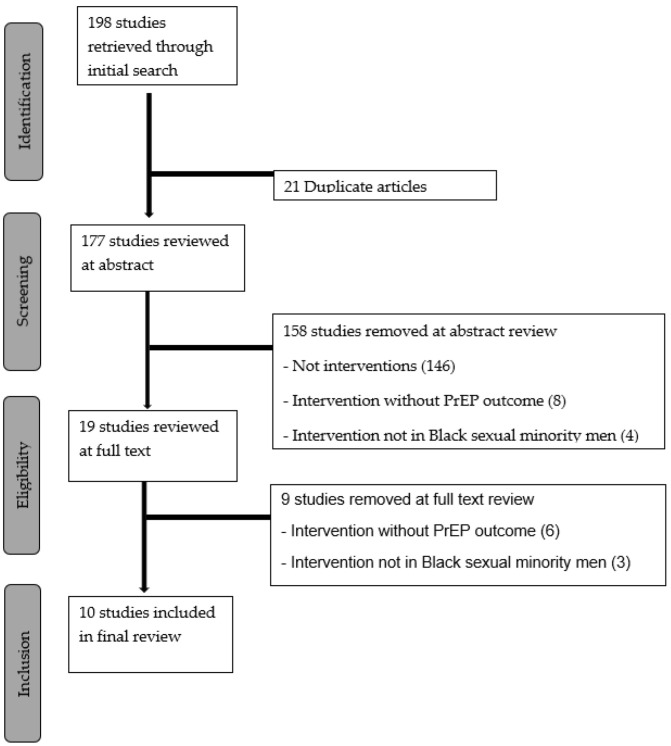
Flow-chart for systematic inclusion of studies according to PRISMA guidelines.

**Table 1 ijerph-19-01934-t001:** Design characteristics of the included interventions (*n* = 10).

Study Title	Country	Intervention Design	Length	Sample Size	Bias Risk
Social network intervention to increase pre-exposure prophylaxis (PrEP); awareness, interest, and use among African American men who have sex with men	U.S.A., Wisconsin	Single group	3 months	37 to 31	Low
Adherence to Pre-exposure Prophylaxis in Black Men Who Have Sex with Men and Transgender Women in a Community Setting in Harlem, NY	U.S.A., New York	Randomized Controlled Trial	12 months	205 to 79	Low
A Pragmatic Randomized Controlled Trial to Increase PrEP Uptake for HIV Prevention: 55 Week Results From PrEP Chicago	U.S.A., Chicago	Randomized Controlled Trial	55 weeks	423 to 342	Low
A Randomized Controlled Pilot Study of a Culturally Tailored Counseling Intervention to Increase Uptake of HIV Pre-exposure Prophylaxis Among Young Black Men Who Have Sex with Men in Washington, DC	U.S.A., Washington, D.C.	Randomized Controlled Trial	3 months	50 to 44	Low
Embedding a Linkage to Preexposure Prophylaxis Care Intervention in Social Network Strategy and Partner Notification Services: Results from a Pilot Randomized Controlled Trial	U.S.A., Chicago	Randomized Controlled Trial	12 months	146	Low
Sex, PrEP, and Stigma: Experiences with HIV Pre-exposure Prophylaxis Among New York City MSM Participating in the HPTN 067/ADAPT Study	U.S.A., New York	Randomized Controlled Trial (Parent Study)	24 weeks (Parent Study)	179 (Substudy)	Low
Predictors of PrEP Uptake Among Patients with Equivalent Access	U.S.A., New York	Non-randomized	12 months	368	Low
Pre-exposure prophylaxis initiation and adherence among Black men who have sex with men (MSM) in three U.S.A. cities: results from the HPTN 073 study	U.S.A., National multisite	Single group	52 weeks	226 to 209	Low
Small Randomized Controlled Trial of the New Passport to Wellness HIV Prevention Intervention for Black Men Who Have Sex with Men (BMSM)	U.S.A., Los Angeles	Randomized Controlled Trial	6 months	105 to 61	Low
Integrated Next-Step Counseling (iNSC) for Sexual Health and PrEP Use Among Young Men Who Have Sex with Men: Implementation and Observations from ATN110/113	U.S.A., National multisite	Single Group (Parent Study)	48 weeks	178	Low

**Table 2 ijerph-19-01934-t002:** Key variables in included interventions (*n* = 10).

Study Title	Sample Size	% PrEP Changes	Intervention Mechanisms and Targets
Social network intervention to increase pre-exposure prophylaxis (PrEP) awareness, interest, and use among African American men who have sex with men	37 to 31	PrEP Use; 3% to 12%; (+9%, OR = 4.41) *	PrEP Knowledge; PrEP Attitudes; PrEP Stigma; Descriptive/Subjective Norms; Self-Efficacy; PrEP Intention; PrEP Willingness; HIV/AIDS Communications; PrEP Communications; Provider Discussions; Peer Navigation
Adherence to Pre-exposure Prophylaxis in Black Men Who Have Sex with Men and Transgender Women in a Community Setting in Harlem, NY	205 to 79	PrEP Adherence; 30% to 30%; (+0%, OR = 1.00)	PrEP Access; Peer Navigation; Group-based Social Support; PrEP Education; General Health Linkage; Case Management; Counseling
A Pragmatic Randomized Controlled Trial to Increase PrEP Uptake for HIV Prevention: 55 Week Results From PrEP Chicago	423 to 342	PrEP Referrals; (OR = 1.50) *	PrEP/HIV Knowledge; PrEP Communications; Group-based Social Support
A Randomized Controlled Pilot Study of a Culturally Tailored Counseling Intervention to Increase Uptake of HIV Pre-exposure Prophylaxis Among Young Black Men Who Have Sex with Men in Washington, DC	50 to 44	PrEP Use (0% to 16%, OR > 9.99) *	PrEP Counseling; PrEP Access; General Health Linkage; HIV Education; Peer navigation
Embedding a Linkage to Preexposure Prophylaxis Care Intervention in Social Network Strategy and Partner Notification Services: Results from a Pilot Randomized Controlled Trial	146	PrEP Initiation (11% to 24%, OR = 2.56) *	PrEP Counseling; PrEP Access; Peer Navigation
Sex, PrEP, and Stigma: Experiences with HIV Pre-exposure Prophylaxis Among New York City MSM Participating in the HPTN 067/ADAPT Study	179 (Substudy)	PrEP use during sex (47% to 66% *, OR = 2.19)	PrEP Counseling; PrEP Access; Peer Navigation; PrEP Stigma; Sexual Stigma
Predictors of PrEP Uptake Among Patients with Equivalent Access	368	PrEP Initiation (72%, Single group)	PrEP Counseling; PrEP Access; General Health Linkage
Pre-exposure prophylaxis initiation and adherence among Black men who have sex with men (MSM) in three U.S.A. cities: results from the HPTN 073 study	226 to 209	PrEP Initiation (79%, Single group), 6-month PrEP use (66%, Single group)	PrEP/HIV Knowledge; PrEP Communications; HIV Risk Counseling
Small Randomized Controlled Trial of the New Passport to Wellness HIV Prevention Intervention for Black Men Who Have Sex with Men (BMSM)	105 to 61	PrEP Use in past 6 months (OR = 3.03)	PrEP/HIV Knowledge; PrEP Communications; HIV/STI Risk Counseling; STI Testing; Employment/Housing Support; Legal Resources; General Health Linkage; Peer Mentoring; Group-based Social Support
Integrated Next-Step Counseling (iNSC) for Sexual Health and PrEP Use Among Young Men Who Have Sex with Men: Implementation and Observations from ATN110/113	178	48-week PrEP Adherence (34%, Single group).	PrEP Counseling; PrEP Access; General Health Linkage; HIV/STI Risk Counseling

* Statistically significant (*p* < 0.05) difference between groups.

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
