# Peer review of "Pre-Exposure Prophylaxis Interventions among Black Sexual Minority Men: A Systematic Literature Review"

_ijerph, 2022, doi:10.3390/ijerph19041934_

Round 1

Reviewer 1 Report

I thought this manuscript was well-written and offered a compelling and helpful literature review about PrEP use among BSMM. Additionally, the finding that the PrEP interventions identified were able to increase PrEP initiation but not PrEP adherence was significant.

I only have some minor suggestions that might enhance the manuscript, but overall I thought the length of the paper was appropriate, the tables were clear, and the writing was concise and straight-forward. I appreciated the clear language and careful editing.

Early in the introduction, the authors state that HIV diagnoses increased from 2010 to 2015. To give readers a more complete picture, they may want to state that since 2015 HIV infections have remained stable (https://www.cdc.gov/hiv/group/bmsm/incidence.html) even though BSMM are still experience significant HIV-related health disparities. I do think it’s important that people realize that HIV incidence is no longer rising among BSMM and improvements have been made.

I thought the Methods section was clearly written and easy for readers to follow.

In Section 3.3 (line 135) the authors state that Table 2 describes a wide range of PrEP-related mechanisms included in the interventions. I suggest describing what they mean by “mechanisms.” In general, I think of mechanisms as intervention strategies or the way in which interventions operate.  The third column of Table 2 seems to combine mechanisms (e.g., peer navigation, counseling, case management, group based social support, access, education, linkage, stigma) with intervention targets or the content of the intervention (e.g., knowledge, attitudes, stigma).  To improve clarity, I might suggest separating the two kinds of variables listed or maybe separate them into different columns (or define mechanisms to be inclusive of both kinds of variables).  Additionally, I think providing more detailed descriptions of some of these variables might be worthwhile. For example, it’s not necessarily clear to me what PrEP Access, Routine Healthcare Access, PrEP Communications, and General Health Linkage actually mean.

In Section 3.4, I’m not sure what “referrals for PrEP” means as an outcome.  Does that mean that a participant accepted a referral to PrEP, actually was seen by a PrEP provider, or something else?

In this sentence “Among those measuring PrEP use/initiation, this included PrEP use/initiation at the end of the study period…”  I’m not sure what “this” is referring to. This what? 

In addition, the one study that defined PrEP uptake among those sexually active is a very interesting them that I think could be expanded upon in the Discussion.  Assessing PrEP Adherence (or drawing conclusions from PrEP adherence data) can be complicated because BSMM may start and stop PrEP use depending on changing level of sexual risk, and not all sexually active men are necessarily at risk for HIV. So, the authors may want to point out that non-adherence (or perhaps not persisting on PrEP over time) may mean that someone is taking PrEP ineffectively or it may mean that their personal circumstances have changed and no longer warrant PrEP use.  Explaining the difference between PrEP adherence and PrEP persistence might also be a topic to explore in the discussion section especially when highlighting areas for future research.

In section 4, lines 174-179 double check to make sure effectiveness and efficacy aren’t being used interchangeably.  Sometimes I thought effective was being used improperly and efficacious might be more accurate.

Author Response

Dear Reviewer,

Thank you for allowing us to submit a revised version of our manuscript “Pre-exposure prophylaxis Interventions among Black Sexual Minority Men: A Systematic Literature Review” to the International Journal of Environmental Research and Public Health. We thank the reviewers for their time and diligence in reviewing this paper. We believe that the suggested revisions will undoubtedly improve our paper. A revised manuscript file includes revisions. Below you will find the reviewers’ original comments in normal text and the authors’ responses in bold. Quoted sections are changes made to the manuscript. In addition to the recommended changes, we have also made minor updates for clarity throughout the paper. We appreciate the reviewer recommendations and believe we have thoughtfully incorporated this helpful guidance.

Reviewer 1

I thought this manuscript was well-written and offered a compelling and helpful literature review about PrEP use among BSMM. Additionally, the finding that the PrEP interventions identified were able to increase PrEP initiation but not PrEP adherence was significant. I only have some minor suggestions that might enhance the manuscript, but overall I thought the length of the paper was appropriate, the tables were clear, and the writing was concise and straight-forward. I appreciated the clear language and careful editing.

Thank you.

Early in the introduction, the authors state that HIV diagnoses increased from 2010 to 2015. To give readers a more complete picture, they may want to state that since 2015 HIV infections have remained stable (https://www.cdc.gov/hiv/group/bmsm/incidence.html) even though BSMM are still experiencing significant HIV-related health disparities. I do think it’s important that people realize that HIV incidence is no longer rising among BSMM and improvements have been made.

We agree with the reviewers’ suggestion about mentioning more recent HIV incidence trends and have revised the manuscript. The revised text reads as follows:

 “Additionally, HIV diagnoses have increased among young adult BSMM (aged 25 to 34) from 2010 to 2015 [1]. Although incidence rates among BSMM have remained stable since 2015, it is estimated that if current incidence persists, one in two BSMM will be diagnosed with HIV in their lifetime [2].”

I thought the Methods section was clearly written and easy for readers to follow.

Thank you.

In Section 3.3 (line 135) the authors state that Table 2 describes a wide range of PrEP-related mechanisms included in the interventions. I suggest describing what they mean by “mechanisms.” In general, I think of mechanisms as intervention strategies or the way in which interventions operate.

We thank the reviewer for bringing attention to this. We agree with the reviewer that “mechanisms” may not be clear enough to readers. To avoid confusion, we have changed this to “Mechanisms and Targets” to better reflect that this incudes mechanisms and targets as described in the point below.

The third column of Table 2 seems to combine mechanisms (e.g., peer navigation, counseling, case management, group based social support, access, education, linkage, stigma) with intervention targets or the content of the intervention (e.g., knowledge, attitudes, stigma). To improve clarity, I might suggest separating the two kinds of variables listed or maybe separate them into different columns (or define mechanisms to be inclusive of both kinds of variables).

We have changed the third column of Table 2 to be “Intervention Mechanisms and Targets.” We feel that this change would best describe the list of key variables within each intervention study and would thus not require the addition of an additional column.

Additionally, I think providing more detailed descriptions of some of these variables might be worthwhile. For example, it’s not necessarily clear to me what PrEP Access, Routine Healthcare Access, PrEP Communications, and General Health Linkage actually mean.

We have revised the results to describe each of these in more detail (Section 3.3):

“Every single intervention included PrEP access (by providing PrEP through the study) and PrEP counseling (discussing the process of starting and maintaining PrEP, reasons for starting PrEP, and answering participant questions) as a component of the intervention. The majority of interventions (7 out of 10) also included a peer-related component, such as a peer navigator, or peer-based support groups. Improving PrEP communications (e.g., willingness to discuss PrEP with peers, family, and healthcare workers) was a common focal point. Other commonly utilized intervention targets were PrEP knowledge, HIV knowledge, and HIV/STI risk counseling. Half of the interventions included linkage to general healthcare as a component (e.g., linkage to over-all routine healthcare services not limited or related to PrEP).”

We have changed Routine Healthcare Access to General Health Linkage, as the latter better captures what we intended. We have also changed PrEP Conversations to PrEP Communications for the same reason.

In Section 3.4, I’m not sure what “referrals for PrEP” means as an outcome. Does that mean that a participant accepted a referral to PrEP, actually was seen by a PrEP provider, or something else?

We have revised to clarify as follows:

“There is one study mentioned in our review using PrEP referrals as an outcome. Here this included referrals to PrEP care from a dedicated PrEP referral line and attendance of a first referred PrEP care clinic appointment.”

In this sentence “Among those measuring PrEP use/initiation, this included PrEP use/initiation at the end of the study period…”  I’m not sure what “this” is referring to. This what?

We have revised the text to read as follows: ”Among the studies measuring PrEP use/initiation at the end of the study period, one study measured PrEP use while sexually active.”

Discussion: In addition, the one study that defined PrEP uptake among those sexually active is a very interesting theme that I think could be expanded upon in the Discussion.

We have revised the text to expand upon this as follows:

“One study captured some of these personal circumstances related to PrEP use, as the out-come measured was PrEP use while sexually active. This provides additional context that helps us more effectively interpret reasons for inconsistency in PrEP use. Future research should investigate this further.”

Assessing PrEP Adherence (or drawing conclusions from PrEP adherence data) can be complicated because BSMM may start and stop PrEP use depending on changing level of sexual risk, and not all sexually active men are necessarily at risk for HIV. So, the authors may want to point out that non-adherence (or perhaps not persisting on PrEP over time) may mean that someone is taking PrEP ineffectively or it may mean that their personal circumstances have changed and no longer warrant PrEP use. Explaining the difference between PrEP adherence and PrEP persistence might also be a topic to explore in the discussion section especially when highlighting areas for future research.

We agree, this is an excellent point. We have added this to the discussion (2nd paragraph, last 2 sentences) as follows:

“Interpretations of non-adherence to PrEP can be challenging however, as this could reflect that someone is taking PrEP ineffectively, or that their personal circumstances no longer warrant PrEP use. Identifying the reason for PrEP cessation is especially important for determining the efficacy of PrEP interventions.”

In section 4, lines 174-179 double check to make sure effectiveness and efficacy aren’t being used interchangeably. Sometimes I thought effective was being used improperly and efficacious might be more accurate.

Thank you for this comment. The revised text reads as follows:

“Peer-based interventions were frequently utilized, and these were a largely efficacious approach to promoting PrEP initiation among BSMM. Overall, focusing on these constructs and utilizing peer-related components appears to have demonstrated promise in increasing PrEP initiation among BSMM, though this efficacy was not observed using adherence outcomes.”

Reviewer 2 Report

Pre-exposure prophylaxis Interventions among Black Sexual 2 Minority Men: A Systematic Literature Review

This systematic literature review aims to identify characteristics of successful interventions between 2016 and 2021 that promote PrEP use among BSMM. It is an important topic among a population that remains heavily impacted by HIV in the US. The manuscript could benefit from the following revisions:

Intro: Update HIV prevalence estimates (there are more recent estimates than 2015).

The first three paragraphs need to be re-written for clarity. Line 42-47 has incomplete sentences & is difficult to follow.

Methods:

Inclusion criteria: Move outcomes to outcomes section. Also, there is a discrepancy in the outcomes listed in line 88 (PrEP initiation or adherence) an in the outcome section (PrEP uptake). Please clarify.

Bias assessment: Line 103 is confusing. Quasi-experimental designs can still have a control group, so the use of the MASTARI is unclear.  

Results:

Why were articles excluded based on the abstract? Please list as this is a significant number of articles.

Lines 155-157 break into multiple sentences.

Table 2. It would be helpful to have the sample size of each study here as well.

Author Response

Dear Reviewer,

Thank you for allowing us to submit a revised version of our manuscript “Pre-exposure prophylaxis Interventions among Black Sexual Minority Men: A Systematic Literature Review” to the International Journal of Environmental Research and Public Health. We thank the reviewers for their time and diligence in reviewing this paper. We believe that the suggested revisions will undoubtedly improve our paper. A revised manuscript file includes revisions. Below you will find the reviewers’ original comments in normal text and the authors’ responses in bold. Quoted sections are changes made to the manuscript. In addition to the recommended changes, we have also made minor updates for clarity throughout the paper. We appreciate the reviewer recommendations and believe we have thoughtfully incorporated this helpful guidance.

Reviewer 2

This systematic literature review aims to identify characteristics of successful interventions between 2016 and 2021 that promote PrEP use among BSMM. It is an important topic among a population that remains heavily impacted by HIV in the US. The manuscript could benefit from the following revisions:

Intro: Update HIV prevalence estimates (there are more recent estimates than 2015).

Thank you. We have updated to cite two more recent CDC reports, one from 2018 and one from 2020. We have also revised the text as follows:

“Additionally, HIV diagnoses have increased among young adult BSMM (aged 25 to 34) from 2010 to 2015 [1]. Although incidence rates among BSMM have remained stable since 2015, it is estimated that if current incidence persists, one in two BSMM will be diagnosed with HIV in their lifetime [2].”

The first three paragraphs need to be re-written for clarity. Line 42-47 has incomplete sentences & is difficult to follow.

We have revised the introductory paragraphs for better clarity, and re-worded lines 42-47 to convey more clearly that it is a list of factors, as follows:

“Barriers to PrEP utilization may include individual factors as well as social and structural barriers. Previous research has identified the following barriers to HIV prevention: difficulties accessing healthcare (e.g. a lack of health insurance, geographic distance), infrequent routine healthcare utilization, socioeconomic difficulties, low self-perceived HIV risk, depressive symptoms, and HIV stigma [3, 5, 6, 8-11].”

Methods:

Inclusion criteria: Move outcomes to outcomes section.

We have moved the outcomes to the outcomes section below, as follows:

“The outcomes measured for all studies included PrEP initiation, PrEP referrals, and PrEP adherence.”

Also, there is a discrepancy in the outcomes listed in line 88 (PrEP initiation or adherence) an in the outcome section (PrEP uptake). Please clarify.

We have revised to clarify as follows:

“We included studies in peer-reviewed journals that used an intervention design, measured PrEP initiation, referral, or adherence as an outcome of the intervention, and were conducted among a sample consisting of more than 50% Black sexual minority men.”

Bias assessment: Line 103 is confusing. Quasi-experimental designs can still have a control group, so the use of the MASTARI is unclear.

The JBI-MASTARI has separate instruments for both experimental and quasi-experimental designs. We have revised to correct our original sentence as follows:

“We used the JBI-MASTARI instruments for randomized controlled trials and quasi-experimental studies.”

Results:

Why were articles excluded based on the abstract? Please list as this is a significant number of articles.

We have added the following to clarify this:

“Of a starting total 198 articles, 21 were excluded as duplicates, and 158 were excluded based on the abstract (Figure 1). Articles were excluded based on the abstract if it was evident that they were not interventions focused on PrEP.”

Lines 155-157 break into multiple sentences.

We have split this, and an adjacent large sentence, into two sentences as follows:

“Odds ratios were only included for multi-group interventions or interventions with pre-test and post-test PrEP measures. Odds ratios for single group interventions restricted to those not using PrEP at baseline could not be calculated.”

“The four interventions with the largest effect estimates (OR>2.00) all included a peer navigation component, though the one intervention with a null effect estimate did as well. This latter study utilized PrEP adherence however, while the former studies all measured PrEP initiation.”

Table 2. It would be helpful to have the sample size of each study here as well.

We have added the sample size to the table.

Round 2

Reviewer 2 Report

All concerns were adequately addressed.